# Multi-responsive chitosan-based hydrogels for controlled release of vincristine

Bahareh Farasati Far [1], Mohsen Omrani[1], Mohammad Reza Naimi Jamal [1✉] & Shahrzad Javanshir [2✉]

As medical research progresses, the derivation and development of biological materials such as hydrogels have steadily gained more interest. The biocompatibility and non-toxicity of chitosan make chitosan hydrogels potential carriers for drug delivery. This work aims to develop two multi-reactive, safe, and highly swellable bio-hydrogels consisting of chitosan-graft-glycerol (CS-g-gly) and carboxymethyl chitosan-graft-glycerol (CMCS-g-gly), for sustained and controlled drug release, improved bioavailability along with entrapment in nano-carriers, which reduces side effects of vincristine sulphate. CS-g-gly and CMCS-g-gly are successfully prepared and fully characterized using analytical techniques. Under various conditions, the prepared hydrogels exhibit a high swelling ratio. Vincristine-loaded CS-g-gly (VCR/CS-g-gly), and CMCS-g-gly (VCR/CMCS-g-gly) show high encapsulation efficiency between 72.28-89.97%, and 56.97-71.91%, respectively. VCR/CS-g-gly show a sustained release behavior, and the maximum release of VCR from hydrogels reached 82% after 120 h of incubation. MCF-7 (breast cancer cell line) and MCF-10 (normal breast cell line) are evaluated for cell viability and apoptosis induction. The in-vitro anti-tumor efficacy is investigated using flow cytometry. The tetrazolium-based MTT assay of hydrogels shows no evidence of significant cytotoxicity in MCF-7 and MCF-10 cells. According to these findings, these hydrogels can effectively deliver drugs to MCF-7 and other breast cancer cells.

[1] Research Laboratory of Green Organic Synthesis & Polymers, Department of Chemistry, Iran University of Science and Technology, 16846-13114 Tehran, Iran. [2] Pharmaceutical and Heterocyclic Compounds Research Laboratory, Department of Chemistry, Iran University of Science and Technology, 16846-13114 Tehran, Iran. ✉email: naimi@iust.ac.ir; shjavan@iust.ac.ir

Based on the World Health 'Organization's mortality report, cancer is the primary cause of mortality in many parts of the world[1]. Cancer cells can proliferate aggressively, infecting surrounding tissues and producing metastasis. Over the past twenty years, advances in chemotherapy have led to a considerable improvement in cancer treatments, with an increase in the cure rate and survival times for cancers at the metastatic stage. Nevertheless, failures remain which raise the question of stopping chemotherapy sooner or later. The main challenges may include the inability of drug to penetrate tumour cells, its toxicity to healthy cells, and its instability in the blood circulating system[2–4]. It is important to evaluate the tumor location while using chemotherapeutic drugs to treat malignant tumours. Interstitial chemotherapy may help mitigate systemic toxicity while still meeting the targets of target therapy[5,6]. Vincristine (VCR) is a vinca alkaloid derived from the Catharanthus roseus plant[7]. It is a highly effective chemotherapeutic agent currently being used to treat acute lymphocytic leukemia, 'Hodgkin's and non-lymphoma, and solid tumors such as sarcomas, neuroblastoma and breast cancer[8]. However, its therapeutic applications are limited due to neurotoxicity and other potential side effects[9]. VCR interacts with and triggers apoptosis in tubulin-expressing cells[10]. In vivo studies indicate that prolonged VCR exposure above a critical threshold dose resulted in more severe cytotoxicity. As a result, the concept of providing VCR-controlled release formulations becomes desirable. Vincristine cannot distinguish between healthy and malignant cells and will attack any dividing cell without discrimination[11]. Multiple rounds of treatment are required to ensure that malignant cells are constantly exposed to vincristine. Naturally, extended exposure to a high dose of the drugs results in adverse side effects[12].

In the last few years, there have been numerous efforts to develop a new VCR formulation with fewer adverse effects and greater therapeutic value. Liposomes, nanoparticles, micelles, and microspheres are only some of the drug delivery platforms used for this purpose, with varied but generally encouraging outcomes[13,14]. Liu et al. synthesized VCR-Au conjugated nanoparticles entrapped in liposome, enhancing anti-tumor efficiency. They investigated the release rate, cytotoxicity, and cytoplasmic delivery in HeLa cells. The new 'formulation's anti-tumor effectiveness and biodistribution were investigated in tumor-bearing mice. Their findings indicated that this novel formulation had stronger anti-cancer activity in mice and fewer adverse effects than free vincristine sulphate[15]. In another work, Aboutaleb and Dinarvand developed a combination of VCR-dextran-loaded solid lipid nanoparticles for brain drug delivery. Their findings indicated that the solid lipid nanoparticles put into the animals' plasma concentrations increased sharply, and their brain drug levels increased quickly[16]. Salar et al. described the synthesis of vincristine-loaded folic acid–chitosan attached nanoparticles through ionic gelation. The spectrophotometric measurement revealed maximum encapsulation effectiveness of 81.25% and a loading capacity of 10.31%, respectively. In vitro, drug release studies revealed that vincristine is released slowly and continuously in phosphate-buffered saline at a pH of 6.7[17]. Chen et al. described a method for preparing vincristine sulfate (VCR) loaded poly(lactide-co-glycolide) microspheres using a water-in-oil-in-oil double-emulsion solvent/evaporation method, followed by the formation of polymeric films via lyophilization of VCR microspheres followed by hydration in a collagen and chitosan swelling solution. 0.3% glutaraldehyde was used to crosslink the films. The film has released VCR over a lengthy period, and the its first burst release was less significant. The film containing chitosan decomposed more slowly than the film missing chitosan in the degradation experiment[18].

Hydrogels have gained the most attention of any biomaterials discovered in recent decades on the basis of their superior biodegradability, biocompatibility, and responsiveness[19]. A hydrogel is a three-dimensional molecule made up of soft, crosslinked components which are non-water soluble. Physical and chemical hydrogels are the two main hydrogels that existing methods can currently be made[20]. Chemical interactions such as ionic crosslinking, hydrogen bonding, and hydrophobic interactions define physical hydrogels[21]. In contrast, covalent bonds between constituent molecules identify as chemical hydrogels[22]. On the other hand, chemical hydrogels are impermeable to water molecules. Large and small compounds can be loaded into hydrogels easily. Hydrogels can react appropriately in the body, physiologically equivalent to other tissues, to receive and distribute nutrients[23].

Deacetylation of chitin produces a non-toxic polysaccharide known as chitosan (CS), approved by the FDA[24]. CS derivatives are commonly used due to their unique properties like biocompatibility, mucoadhesion, non-toxicity and capability to form gels. Carboxymethyl chitosan (CMCS) has appeared as an auspicious biopolymer for the development of new drug delivery systems. CMCS an amphiprotic ether derived from chitosan, exhibit improved aqueous solubility, outstanding biocompatibility, biodegradability, osteogenesis ability and numerous other outstanding physicochemical and biological properties. More stupendously, it can load hydrophobic drugs and displays strong bioactivity which highlight its suitability and extensive usage for preparing different drug delivery systems. Although chitosan is insoluble in water at neutral pH, its amine groups can be protonated to make it soluble in acidic aqueous solutions. Chitosan stays dissolved until the pH drops below 6.5. When chitosan solutions are zneutralized with a strong base up to pH > 6.5, an instantaneous hydrated gel-like precipitate forms, this process removes the positive charge from the chitosan amine groups[25]. Their repulsion is suppressed, resulting in the formation of chitosan–chitosan hydrogen bonds via the amine ($-NH_2$), hydroxyl ($-OH$), carbonyl ($-CO$) and, amide ($-NH-C$) groups[26]. The utilization of polyelectrolyte complexes and various activating agents for altering chitosan for enzyme immobilization applications has received attention[27,28]. As a result of their superior biodegradability and environmental friendliness, natural polymers such as chitosan have recently gained a great deal of attention for immobilization[29]. Multiple interactions among chitosan, glycerol, and formaldehyde may be involved in the gelation mechanism of CS hydrogel[30]. The molecular weight, degree of deacetylation, and concentration of chitosan affect the hydrogel solution behaviour, viscoelasticity and physicochemical characteristics, as well as the gelation procedure and the type and concentration of aldehyde[31,32].

This study aimed to design and characterize a novel synthesis process of the biodegradable, cost-effective, non-toxic, and highly swellable hydrogels and investigate the controlled release and cytotoxicity studies of vincristine sulfate (VCR) anti-cancer drug from hydrogels. VCR is encapsulated within chitosan-graft-glycerol (CS-g-gly) and carboxymethyl chitosan-graft-glycerol (CMCS-g-gly) hydrogels to demonstrate that the drug-hydrogel system is a promising site-specific local delivery vehicle. CS-g-gly and CMCS-g-gly were successfully prepared and fully characterized using Fourier transform infrared (FT-IR) spectroscopy, Proton nuclear magnetic resonance ($^1$HNMR), field-emission scanning electron microscope (FE-SEM), atomic force microscopy (AFM), thermogravimetric analysis (TGA), dynamic light scattering (DLS), and zeta potential techniques. Hydrogel equilibrium water content (EWC) and swelling ratio were evaluated. VCR was loaded into CS-g-gly and CMCS-g-gly hydrogels, and its release profile was evaluated. Cell viability and induction of apoptosis were assessed in MCF-7 and MCF-10 cells using MTT assay and Flow cytometry.

**Fig. 1 Synthesis pathway of hydrogels. a** CS-g-gly, (**b**) CMCS-g-gly.

## Results and discussion

**Synthesis mechanism of CS-based hydrogels.** Various formulations have been used to create chitosan hydrogels. The networking has been achieved via non-covalent physical interactions or covalently crosslinked chemical bindings. The hypothesized mechanisms of glycerol immobilization on CS and CMCS via physical adsorption and covalent attachment, respectively, are depicted in Figs. 1 and 2. At pH < 6.5, CS contains amine groups, so they are protonated in acidic conditions ($NH_3^+$) to facilitate dissolution[33]. CS was activated with formaldehyde by covalent attachment to integrate aldehyde groups on the support surface by interacting with the amino groups to provide enough functional groups for covalent bonding (Fig. 1). Chemical activation resulted in bonds between the free aldehyde groups of CS and amino groups ($NH_2$). A nucleophile (glycerol) attacked iminium ions, forming the CS-g-gly. Hydrogen bonding between chitosan $NH_3^+$ groups and glycerol oxygens is the effective interaction responsible for the sol-gel transition. A solution of ammonia was then added to this mixture in order to obtain a gel, which in this

paper is referred to CS-g-gly nanohydrogels. In order to synthesize CMCS-g-gly, carboxymethyl chitosan has been utilized in place of CS. Hydrogen bonding, electrostatic attraction, as well as hydrophobic interaction,, were found to be the primary mechanisms of gelation. Furthermore, glycerol contributed to the enhancement of the drug-loading efficiency of the nanohydrogels. The drug can get trapped between the ionic bonds of CS and glycerol during the physical adsorption phase. Also, the drug may get trapped in the interfacial gaps generated by the polymer and the formaldehyde[34] (Fig. 2). Additionally, it has been demonstrated that certain polyol phosphate salts can alter the precipitation behavior of chitosan at high pH. When combined with polyol phosphate salts such as STPP, chitosan can generate a pH-dependent and thermosensitive hydrogel. Spectroscopic methods were used to characterize the products.

## Characterization of hydrogels

*FTIR analysis.* As shown in Fig. 3a, the strong broad peak at 3353 cm$^{-1}$ in chitosan could be due to the stretching vibration of

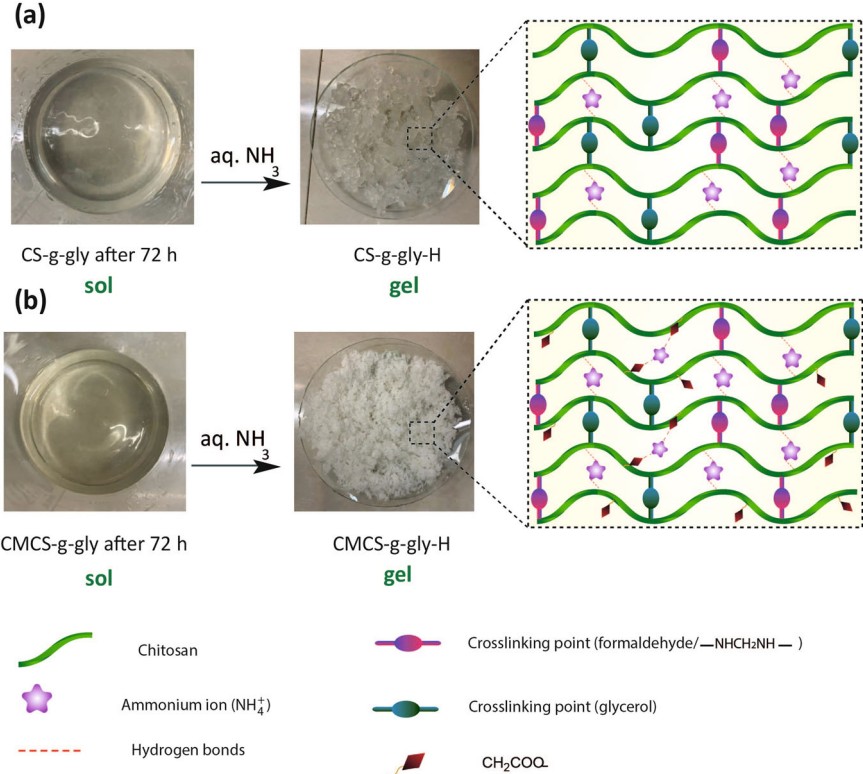

**Fig. 2 Design strategy for CS-g-gly and CMCS-g-gly hydrogels. a** Gelling of CS and (**b**) gelling of CMCS-g-gly with NH4 + binder after 48 h.

–OH. The adsorptions in the area of 1655 and 1590 cm$^{-1}$ could also be attributed to the amide groups. The stretching frequency of the methylene group is observed at 2867 cm$^{-1}$. The FTIR spectrum of glycerol is given in Fig. 3b. The broad peak at 3275 cm$^{-1}$ is due to the –OH groups and the one at 2880-2933 cm$^{-1}$ is that of the –CH group. The strong peak at around 1415 cm$^{-1}$ could be attributed to the bending of the C–O–H of glycerol[35]. Fig. 3c show the IR spectra of CS-g-gly. The strong peak at around 1403 cm$^{-1}$ could be due to the crosslinking of glycerol to chitosan (C–O–R). Fig. 3d, e show the Na-salt IR spectra of CMCS and CMCS-g-gly. Comparatively to the peaks of chitosan, the presence of peaks at 1616 and 1506 cm$^{-1}$ correspond to the carboxy groups (which overlap with the N-H bend) and -CH$_2$COOH groups (Fig. 3d) suggests that both chitosan amino and hydroxyl groups are carboxymethylated[36]. In Fig. 3e, the strong peak at around 1400 cm$^{-1}$ could be attributed to the crosslinking of glycerol to CMCS (C–O–R).

*$^1$H-NMR.* The $^1$HNMR spectrum of CS-g-gly in D$_2$O is shown in Fig. 4a. The signal at 1.78 ppm can be attributed to the acetyl group (–CH$_3$), whilst the signal at 2.83 ppm corresponds to the H2 proton of CS. The chemical shifts between 3.42 and 4.00 ppm are attributed to the H3–H6 and H10, 11 protons of the chitosan and glycerol. The signal at 4.57–4.66 ppm can be attributed to the methylene groups formed by crosslinking glycerol and chitosan (H8, H9). The hydrogen bonded to the anomeric carbon (H1) gives rise to a signal in the range of 4.96 ppm. The 500 MHz $^1$HNMR spectrum of CMCS-g-gly in D$_2$O is shown in Fig. 4b. The r signal from N–CH$_2$–COO$^-$ and O–CH$_2$–COO$^-$ groups are detected at 3.25 and about 4.60, respectively. Similar peaks are found for CS-g-gly[37].

*FESEM morphology of the hydrogels.* The morphology of the synthesized hydrogel was measured by FESEM. (Fig. 5). The diameter of the CS-g-gly and CMCS-g-gly (58.44 nm and

76.93 nm, respectively) (Fig. 5a, c) increased to 109.6 and 154.19 nm (Fig. 5b, d) after loading with VCR, respectively. A possible reason for this observation is that the drug is successfully loaded into the porous hydrogels before drying. Therefore, the diameter of the VCR-loaded nanohydrogels was larger than the drug-free nanohydrogels. Since the morphology of the prepared CS-g-gly and CMCS-g-gly loaded with VCR formulations showed that nanohydrogels scales were less than 100 nm, the prepared hydrogels can provide the size requirement for efficient cancer therapy applications.

*DLS and Zeta Potential.* Any suspended particle, macromolecule, or material surface will display zeta potential as a physical feature. Microemulsions, colloidal, and protein solutions can all have their formulas fine-tuned, as well as interactions with surfaces predicted and the optimal creation of surface coatings achieved[38]. An understanding of the zeta potential can facilitate the production of trial formulations more efficiently. Therefore, Zeta potential is attributed to the stability of the carrier[39]. Positive zeta potential indicates that the particles have a positive charge. Therefore, colloids with a higher positive or negative zeta potential are electrically stabilized, whereas colloids with a low zeta potential are subject to flocculation or coagulation. The zeta potential graph is shown in Fig. 6d. A summary of the size and charge properties of hydrogels is presented in Table S1. The zeta potentials of the hydrogels were positive and in the range of +48 to +57 mV (Fig. 6b). The size and zeta potential of the vesicles are positively correlated in terms of their stability and release pattern[40]. Hydrogel stability, half-life, and cellular absorption are all enhanced by their small size and comparatively large positive surface charge, or zeta potential[6].

Figure 6a displays the DLS results for synthesized chitosan-based nanohydrogels, having a typical particle size distribution profile with a mean size of about 84.5 and 87.6 nm for CS-g-gly and CMCS-g-gly, respectively. Previous studies have reported the

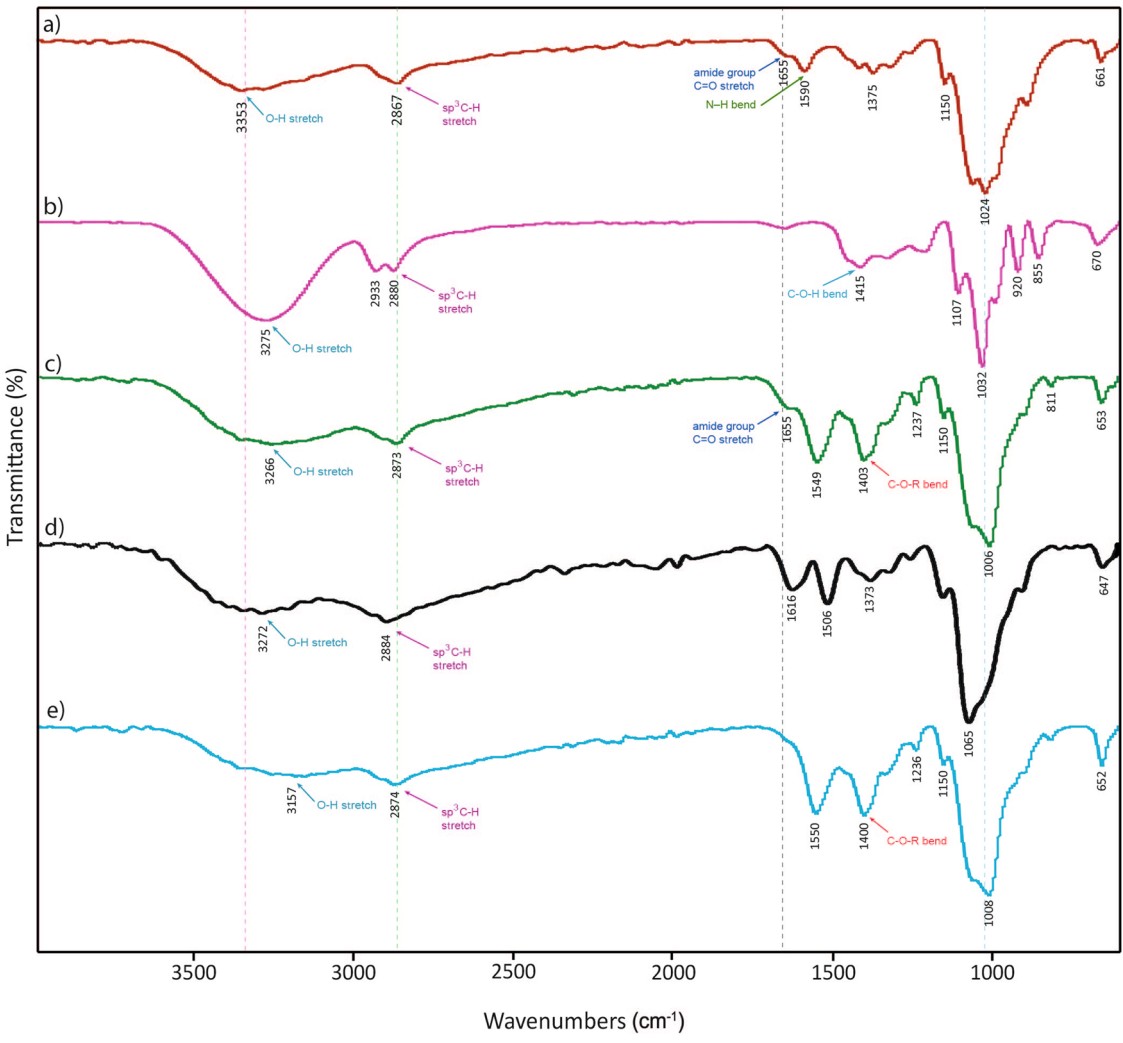

**Fig. 3 FT-IR spectra of the synthesized hydrogels. a** Chitosan, (**b**) Glycerol, (**c**) CS-g-gly, (**d**) CMCS, (**e**) CMCS-g-gly.

size of chitosan-based hydrogels in the range of 200–250 nm[41–43]. Also, Fig. 6a indicates that the nanohydrogels loaded with drugs are 200–300 nm in size. Although the results indicate that drug addiction has increased the mean size of CS nanohydrogels, they are similar to other studies[41,42,44,45]. In comparison with Fig. 6a, when comparing the results of the FESEM and DLS techniques, it is clear that the mean particle sizes estimated by the FESEM technique are smaller. DLS measures hydrodynamic diameter as the particle size. As determined by the DLS technique, particle size may also be overstated due to swelling and aggregation of CS nanohydrogels. As reported by Danaei et al. for the intravenous nanoparticle route of administration, the size should be in the range of 200–2000 nm, which follows our result as an intravenously-use hydrogel[46]. The polydispersity of hydrogels is about 1–3, which shows high polydispersity of hydrogel particles in powder form (Fig. 6c).

*Atomic force microscopy (AFM) of hydrogels.* Compared with DLS, an advantage of AFM is that no mathematical models are required to provide size information. Fig. 7a, b illustrate the AFM images of CS-g-gly and CMCS-g-gly nanohydrogels. The results of AFM confirm that CS-g-gly and CMCS-g-gly hydrogels consist of spherical structures with a diameter consistent with the PCS data. As shown in Fig. 7, CS-g-gly and CMCS-g-gly have a height of around 140 and 50 nm and a width of 80 to 300 nm. AFM was also used in another study to monitor spherical nanogels in which

ovalbumin and ovotransferrin proteins had been incorporated into amphoteric nanogels[47]. AFM measurements were carried out on chitosan and TPP-based nanogels in another study. These nanogels had diameters between 200 and 400 nm[48].

*Thermal Stability Analysis of hydrogels.* In the TGA curve of Cs-g-gly hydrogel, the initial weight loss at 120 °C, which is about 12% weight of the hydrogel, is attributed to the physically adsorbed moisture. The weight loss at 150–230 °C, which was about 5%, is attributed to the glycerine loss. The subsequent mass loss of the sample starts from 230 to 330 °C, which is about 19% of the weight of the polymer and is due to the thermal decomposition of the pyranose ring of chitosan (Fig. 7c). TGA analysis of CMCS-g-gly hydrogels shows about 5% weight loss due to an initial reduction in physically adsorbed water (40 to 150 °C). The weight loss of about 8% at (150–230 °C) is due to the removal of glycerine. The thermal decomposition at 230 to 320 °C, about 25%, can be due to the pyranose ring and carboxyl groups of CMCS (Fig. 7d). The weight loss in the synthesized hydrogels in the range of 330 to 600 °C can be attributed to the further degradation of the polymer structure.

*Investigation of swelling behaviours of multi-responsive hydrogels* The Effect of pH on the swelling ratio of hydrogels.: Figure 8a, d show that CS-g-gly hydrogel showed a nearly comparable swelling ratio in the first 8 h at pH = 5 and 1.2. The differential in

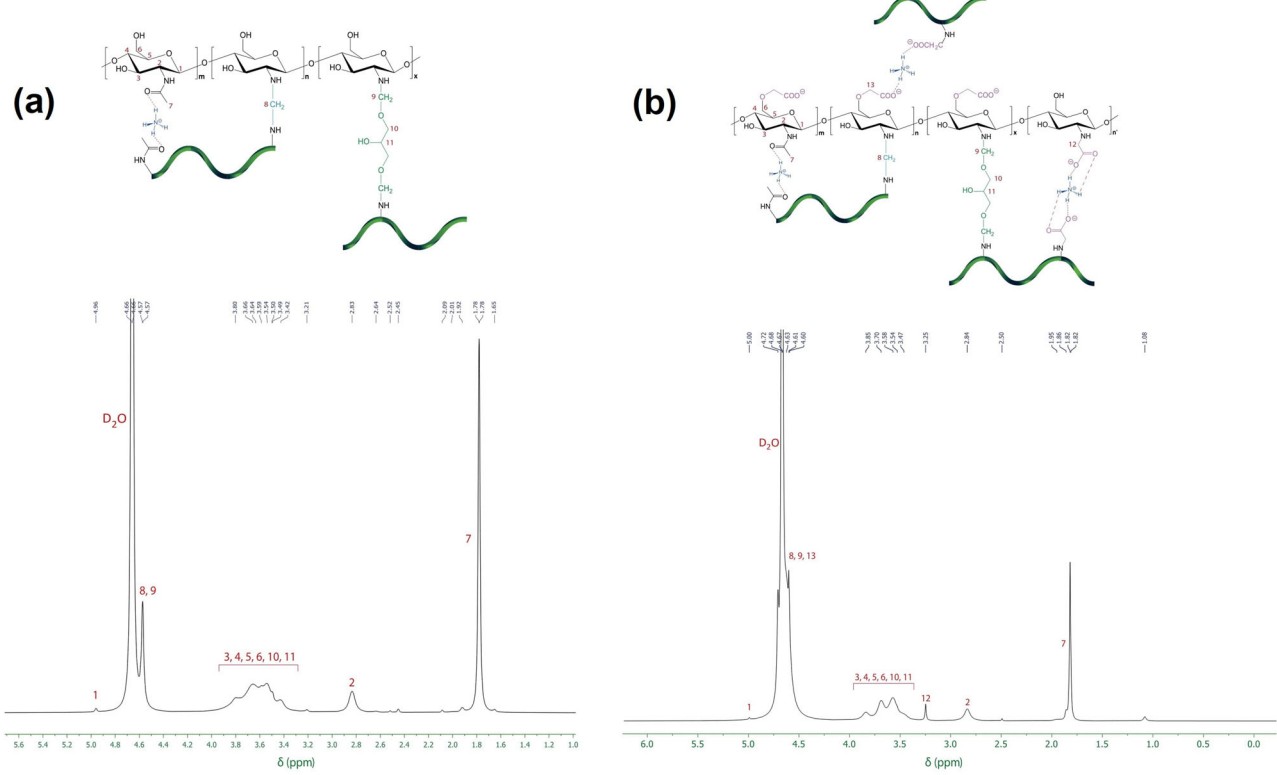

**Fig. 4 ¹HNMR spectra of hydrogels. a** CS-g-gly and (**b**) CMCS-g-gly (500 MHz, D2O, 25 °C).

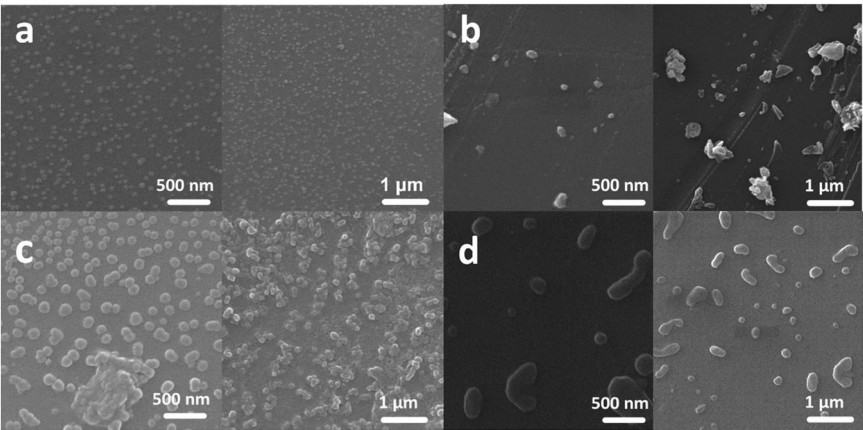

**Fig. 5 SEM morphological characterization of Hydrogels. a** CS-g-gly, (**b**) VCR/CS-g-gly, (**c**) CMCS-g-gly, (**d**) VCR/CMCs-g-gly.

swelling also became more apparent after 24 h, suggesting that the hydrogels are pH sensitive. The CS-g-gly hydrogel, as shown in Fig. 8d, displayed a higher swelling ratio at pH = 5 (which resembles the cancer tissue environment) than pH = 7.4 (which reflects the blood pH). Because the hydrogel contains free –NH₂ groups, they can be transformed into –NH₃⁺ and repulsed by the –NH₃⁺ groups, which causes the CS-g-gly hydrogel to swell in acidic environments. These –NH₃ + groups change to –NH₂ when the pH is neutral or basic, which can lead to a lower swelling ratio. Furthermore, swelling at pH = 7.4 can be explained by the osmotic pressure of the blood. In tumor tissue environments (pH = 5.0, Fig. 8d), the swelling ratio is greater than the swelling ratio at pH = 7.4 and also greater than swelling at pH = 1.2 due to its acidic characteristic and osmotic pressure. A maximum swelling ratio is achieved after 24 h, respectively, in the order of pH = 1.2 > 5 > 7.4. The early-hour swelling

differential pattern at pH = 1.2 and 5 is clearly more noticeable than at pH = 7.4. Also, CMCS-g-gly hydrogels showed differential swelling after 24 h, confirming pH sensitivity. The repulsive contact of chains may result in the deprotonation of COOH groups to COO- anions. (Fig. 8b, f). The swelling rate at pH = 7.4 (which reflects the blood pH) was lower than pH = 5 at 24 h.

The effect of Temperature on the swelling of hydrogels: The swelling behavior of hydrogels at various temperature of 5, 25, 37, and 55 °C is also investigated and shown in Fig. 9a–f. It is clear that increasing water temperature from 5 to 40 °C can increase CS-g-gly hydrogel's swelling ratio (Fig. 9a, d). CS-g-gly hydrogel has weaker hydrogen bonds at higher temperatures, allowing water to penetrate the hydrogel and expand swelling easily. In contrast, the hydrogel has stronger hydrogen bonds at lower temperatures, preventing large volumes of water from penetrating

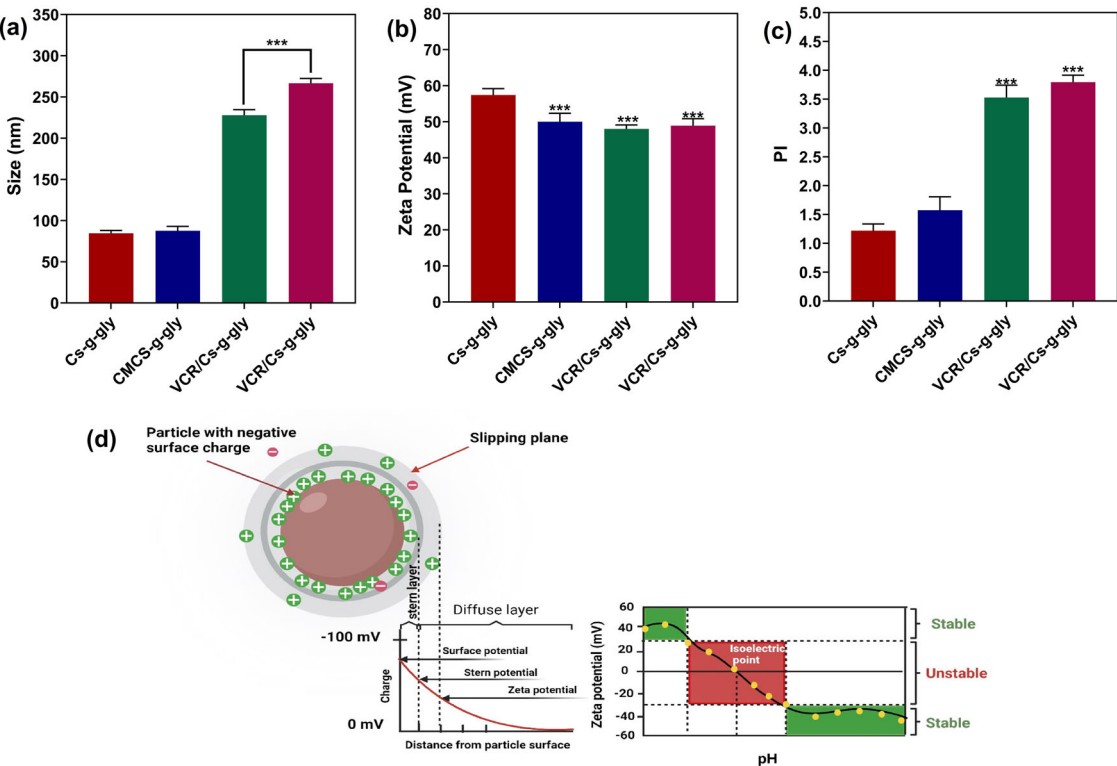

**Fig. 6 Loaded and unloaded hydrogels obtained responses under the optimal conditions. a** hydrogel size, (**b**) Zeta potential, and (**c**) Polydispersity, (**d**) Schematic representation of zeta potential. Data represent means ± standard deviations ($n = 3$). For all charts, ***$p < 0.001$; **$p < 0.01$; *$p < 0.05$.

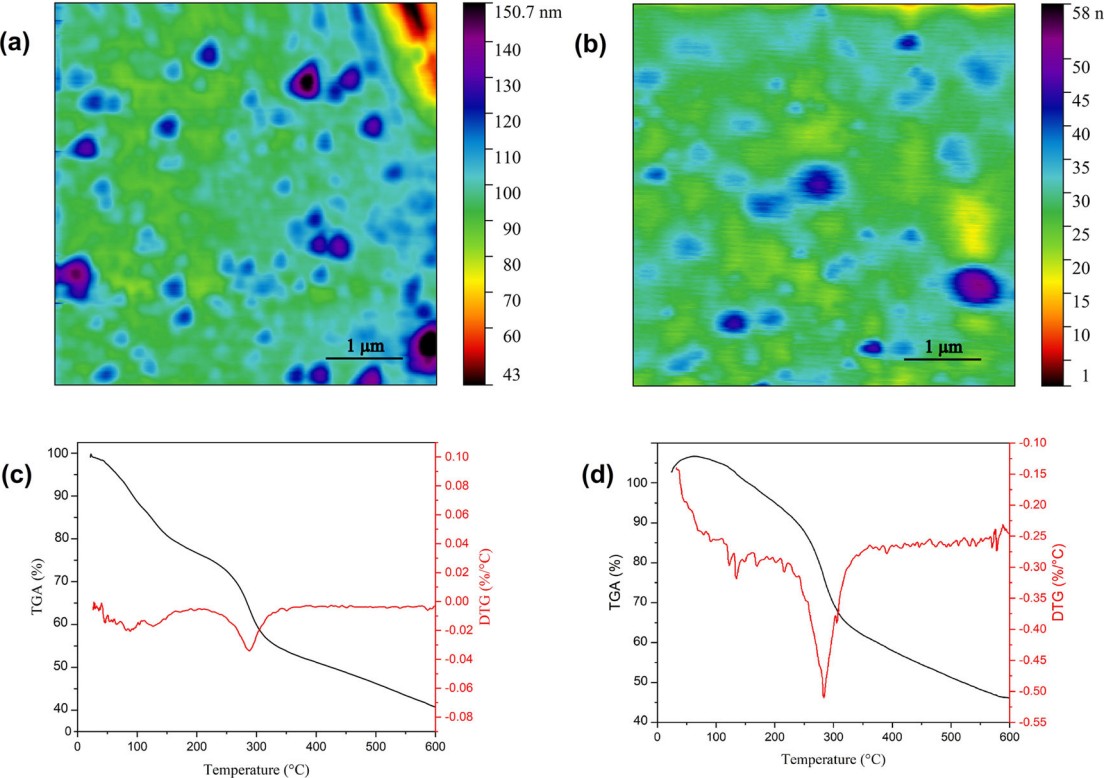

**Fig. 7 Atomic force microscopy (AFM) images. a** CS-g-gly and (**b**) CMCS-g-gly hydrogels showing nanoparticles morphologies, TGA and DTG of (**c**) CS-g-gly and (**d**) CMCS-g-gly hydrogels.

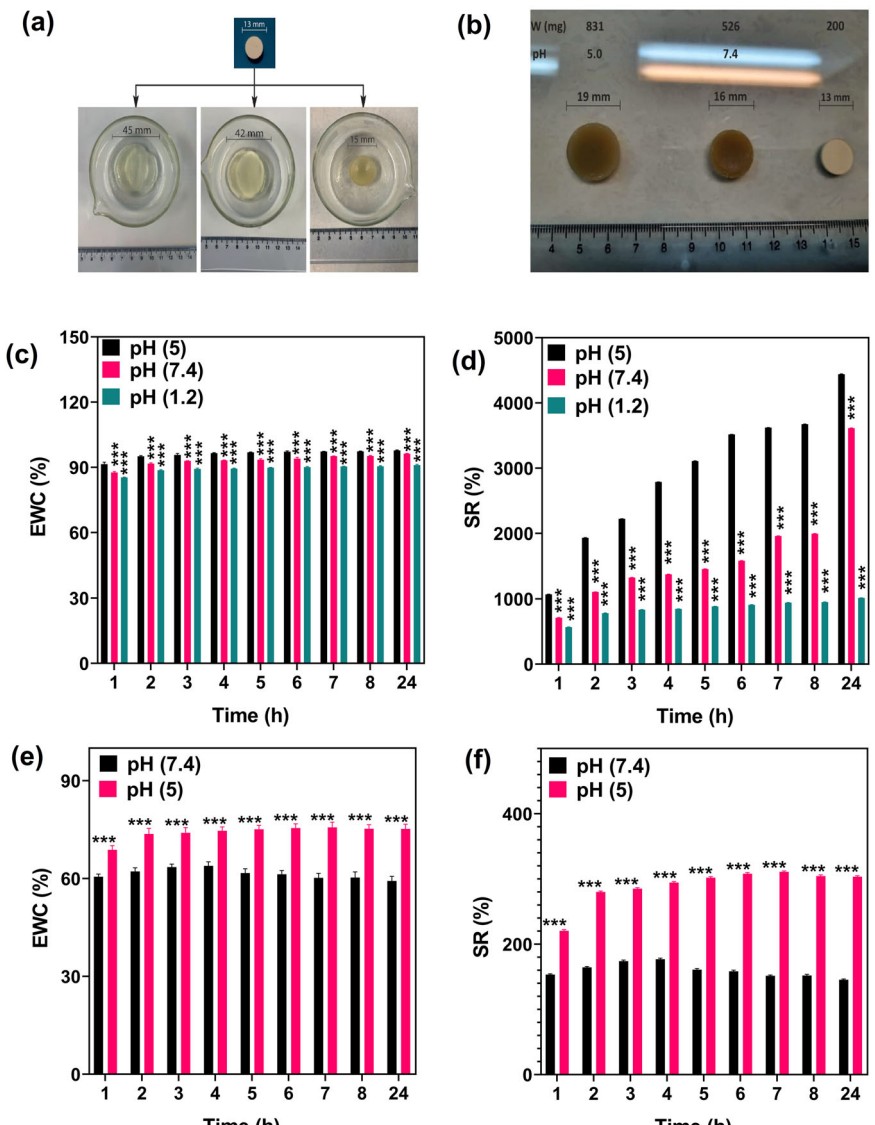

**Fig. 8 The effect of pH on swelling behavior, EWC and SR of hydrogels. a** CS-g-gly dry tablet in pH = 1.2, pH = 5 and pH = 7.4 after 24 h, (**b**) CMCS-g-gly dry tablet in pH = 5 and pH = 7.4 after 24 h, (**c**) CS-g-gly EWC (%), (**d**) CS-g-gly SR (%), (**e**) CMCS-g-gly EWC (%), (**f**) CMCS-g-gly SR (%), in different pH. Each experiment was performed and replicated three times, and the results were displayed using error bars to depict the variance.

the hydrogel and causing it to stiffen up. When the temperature increased between 40 °C and 55 °C, CS-g-gly hydrogel's swelling reduced. Because the hydrogel structure is unstable at higher temperatures (significantly above 37 °C), it is assumed that a part of the hydrogel dissolves in water (55 °C). The swelling of CMCS-g-gly hydrogel in water is depicted in Fig. 9b. The swelling of the CMCS-g-gly hydrogel increases when the temperature increases from 5 to 55 °C. As shown in Fig. 9f, CMCS-g-gly hydrogel exhibits less swelling ratio than CS-g-gly hydrogel. CMCS-g-gly hydrogel possesses more hydrogen bonds and crosslinks (according to Fig. 1) than CS-g-gly hydrogels, resulting in more rigid and robust structures resistant to water penetration. Both hydrogels swell when the temperature increases, which is attributed to weak hydrogen bonding[49,50].

### In vitro drug release study
*Entrapment Efficiency (EE).* VCR was loaded into hydrogels after preparation, and its EE% was studied (Table S2). The results show that increasing STPP percentage decreases EE% and significantly affects the percentage of entrapment efficiency (EE%; $p < 0.05$).

While increasing the carriers: drug ratio had no significant effect on decreasing or increasing EE% (Fig. 10a, b).

*Drug release and kinetic models.* Fig. 10c, d compare the release profile for VCR from the CS-g-gly and CMCS-g-gly hydrogels in PBS at different pH values at 37 °C. The drug release profile of the drug is essential in assessing the effectiveness of a localized delivery system for anti-cancer agents[51]. The pH value also affected the cumulative release profile of VCR from CS-g-gly and CMCS-g-gly hydrogels (Fig. 10c, d). The cumulative release profile of CS-g-gly and CMCS-g-gly was studied in PBS buffer solution at pH = 5 and 7.4. The cumulative release profiles exhibit a similar pattern during the first hours of the experiment at the different pH values. After the initial release, the release rate decreased in the neutral condition while remaining high in the acidic condition. In the first 60 min, 19.05% and 31.50% of the VCR in pH 5 and 9.49% and 4.58% of the VCR in pH 7.4 were released from the CS-g-gly and CMCS-g-gly hydrogels, respectively. The release profile of both hydrogels decreased gradually, possibly due to a decrease in the amount of the drug in the matrix

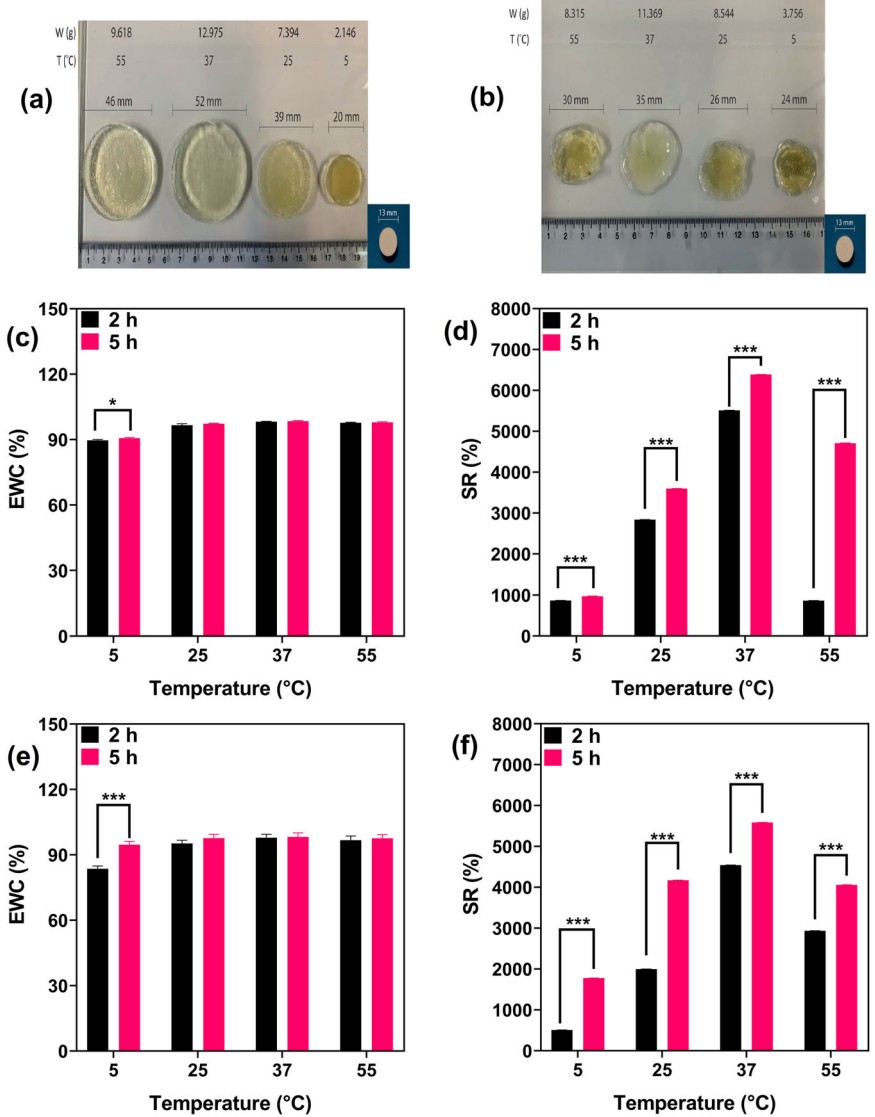

**Fig. 9 The effect of temperature on swelling behavior of hydrogels. a** CS-g-gly dry tablet, (**b**) CMCS-g-gly dry tablet, (**c**) CS-g-gly EWC (%), (**d**) CS-g-gly SR (%) (**e**) CMCS-g-gly EWC (%), (**f**) CMCS-g-gly SR (%), in different temperature after 2 and 5 h. Each experiment was performed and replicated three times, and the results were displayed using error bars to depict the variance.

of chitosan-based hydrogels and a decrease in the concentration gradient of the drug. By the end of 5 days, 82.27% and 75.50% of the VCR in pH = 5 and 25.83% and 5.94% of the VCR in pH = 7.4 were released from the CS-g-gly and CMCS-g-gly hydrogels, respectively. Nearly 100% of the free VCR was released in the PBS solution in the first six hours. Free VCR showed a rapid release pattern with a release rate of about 81.76% in the first four hours. The release of VCR from CS-g-gly and CMCS-g-gly was significantly lower than the free anti-cancer drug, which shows slow and controlled release. When comparing the CS-g-gly hydrogel to the CMCS-g-gly hydrogel, the VCR burst and cumulative release rates were much lower in the CS-g-gly. Glycerol was important in creating both hydrogels through its hydrogen bonding interactions with the chitosan. Drug release was slowed due to the reinforcement of the crosslinked network by hydrogen bonding interactions. A more compact hydrogel structure was formed in CS-g-gly hydrogel, and a significantly slower drug release profile was observed.

These results are consistent with those of other researchers showing that the release profiles change depending on the pH of

the release medium. Pakzad et al.[52] and Ata et al.[53] revealed that the functional group of the model compounds determines how they are released in vitro from chitosan-based hydrogels at varying pH values. This corroborates the findings of Wu et al., who observed that the chitosan hydrogel was pH-sensitive[54]. It was confirmed that CS-based hydrogels swell at low pH levels, forcing the contained drug out with the water. A decrease in the rate of drug diffusion from inside to outside the body was also observed at a pH value of 7.4.

The kinetics of VCR release from the samples were assessed using several different models, including the Korsmeyer–Peppa model, the Higuchi model, the first-order model and the zero-order model[55,56]. The linear curve was determined using the values of the correlation coefficient (r) generated by regression of the plots derived from the models. Since drug release is assumed to be concentration-independent in the zero-order model, drug solubility is necessary for this description. The rate of drug release is described by a first-order rate equation, which is concentration-dependent. In the Higuchi and Korsmeyer–Peppas models, the amount of drug released from a matrix system is proportional to the square root of

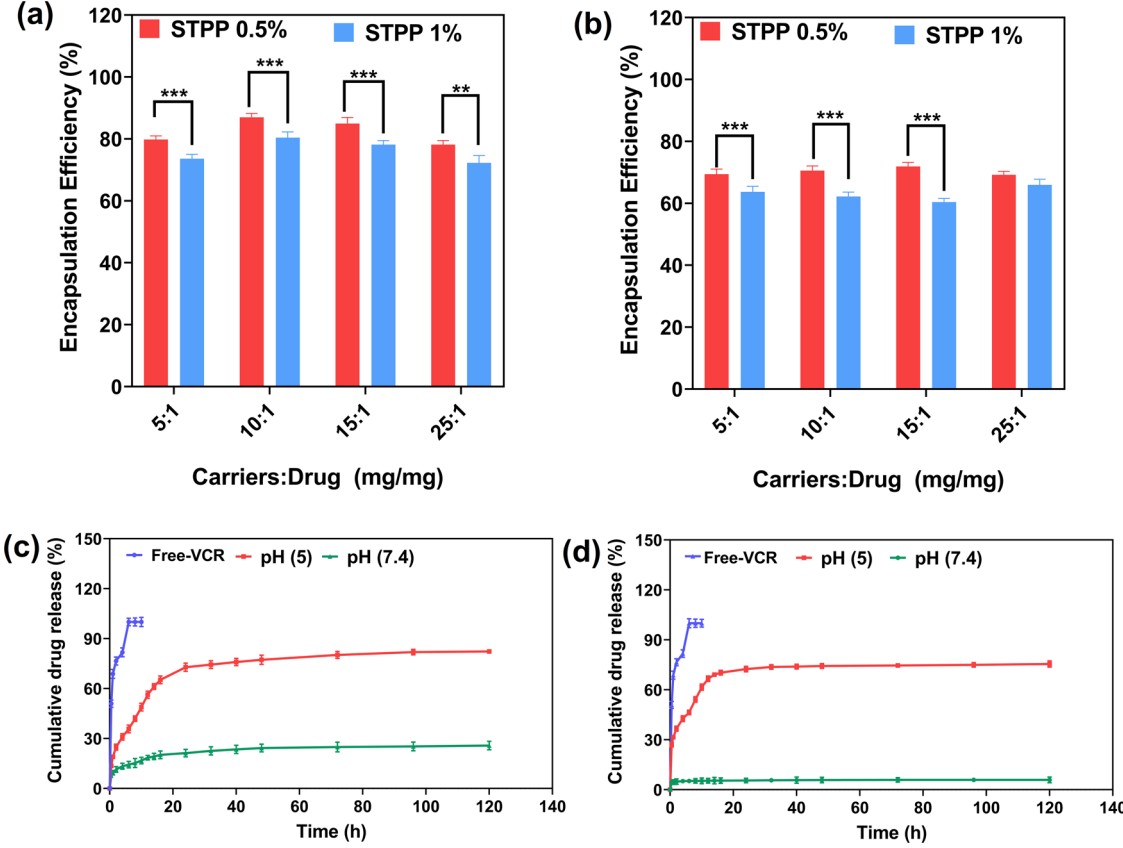

**Fig. 10 Encapsulation efficiency and drug release profile of prepared hydrogels.** Encapsulation efficiency of (**a**) CS-g-gly and (**b**) CMCS-g-gly. Drug release profile of (**c**) VCR/CS-g-gly and (**d**) VCR/CMCS-g-gly and free VCR in PBS (pH 7.4 and 5) after 120 h. Each value represents the mean ± SD of three independent experiments.

time[57,58]. The optimal model for drug release might be determined with a 60% initial release[59]. Korsmeyer–Peppas was the most effective and best-fitting model for release based on the data collected after 24 h. (Table S3).

**Cell viability study**. The MTT method is one of the standard methods for determining the relative cellular metabolic activity. Viability of the human breast cancer cell line MCF-7 and the non-tumorigenic epithelial cell line MCF-10 was measured using the MTT test for both CS-g-gly, CMCS-g-gly, and their drug-loaded hydrogels. The diagram of the absorption of Formazan product produced after dissolving in DMSO for different drug concentrations shows that the percentage of viable cells decreased with increasing doses of vincristine in MCF-7 cells at 24 and 48 h. As shown in Fig. 11a, b, inhibition of tumor cells was lower at low concentrations. In contrast, at higher concentrations (above 25 ng/mL), viability was significantly reduced at 48 h and reached the minimum value of about 30% at 50 ng/mL ($p < 0.0001$). The IC$_{50}$ value in MCF-7 cells in CS-g-gly and CMCS-g-gly loaded with vincristine after 48 h was 22.49 and 24.43, respectively. Generally, cytotoxicity is defined as less than 50% biocompatibility for a given substance. CS-g-gly and CMCS-g-gly drug-free hydrogels did not significantly reduce MCF-7 cell viability compared to drug-loaded hydrogels after 24 h, indicating that they are non-toxic. The further decrease in cell viability after 48 h in CS-g-gly and CMCS-g-gly drug-free hydrogels can represent necrotic cells, as proved by Annexin-V/-FITC/PI Flow cytometry analysis.

MCF-10 non-tumorigenic epithelial cells treated with CS-g-gly and CMCS-g-gly loaded with vincristine hydrogels showed a slight effect on cell viability at different concentrations compared

to MCF-7 cells. As shown in Fig. 11c, d, in the range of 10 ng/mL to 25 ng/mL, including the IC$_{50}$ value of the MCF-7 cell line, the viability of MCF-10 normal cells was higher than 70% at 48 h. Only at the highest concentrations (50 ng/mL) did the drug-loaded hydrogels reduce normal cell viability by nearly 40%. A comparison of the results obtained from the MTT cell viability assay for tumor cell line and non-tumor cell line showed a significant selectivity of the inhibitory effect of vincristine-loaded hydrogels. CS-g-gly and CMCS-g-gly drug-free hydrogels also did not significantly reduce MCF-10 cell viability compared to drug-loaded hydrogels after 48 h, showing that the synthesized hydrogels are non-toxic. Moreover, the MTT heatmap illustrates the correlation between nanohydrogels and the classification of nanohydrogels as potentially suitable carriers after 48 h. Red indicates a lower toxicity level, while purple indicates a higher level of toxicity. The concentration-dependent response of VCR to MCF-10 and MCF-7 cells can be seen using heat map profiles.

**Annexin-V/-FITC/PI Flow cytometry analysis of MCF-7 breast cancer cells**. Annexin-V/-FITC/PI Flow cytometry analysis for CS-g-gly, CMCS-g-gly, CS-g-gly loaded with vincristine, and CMCS-g-gly loaded with vincristine are shown in Fig. 12a, b. MCF-7 cells treated with different samples were labelled with Annexin V-FITC / PI and analyzed by flow cytometry to identify apoptotic cells. As can be seen, the percentage of living cells (Q4 region) in the control group, which is 84%, is higher than in the other groups. CS-g-gly and CMCS-g-gly samples did not affect the induction of apoptosis in the MCF-7 cell line, which indicates that the synthesized hydrogels are non-toxic. CS-g-gly loaded with vincristine, and CMCS-g-gly loaded with vincristine can

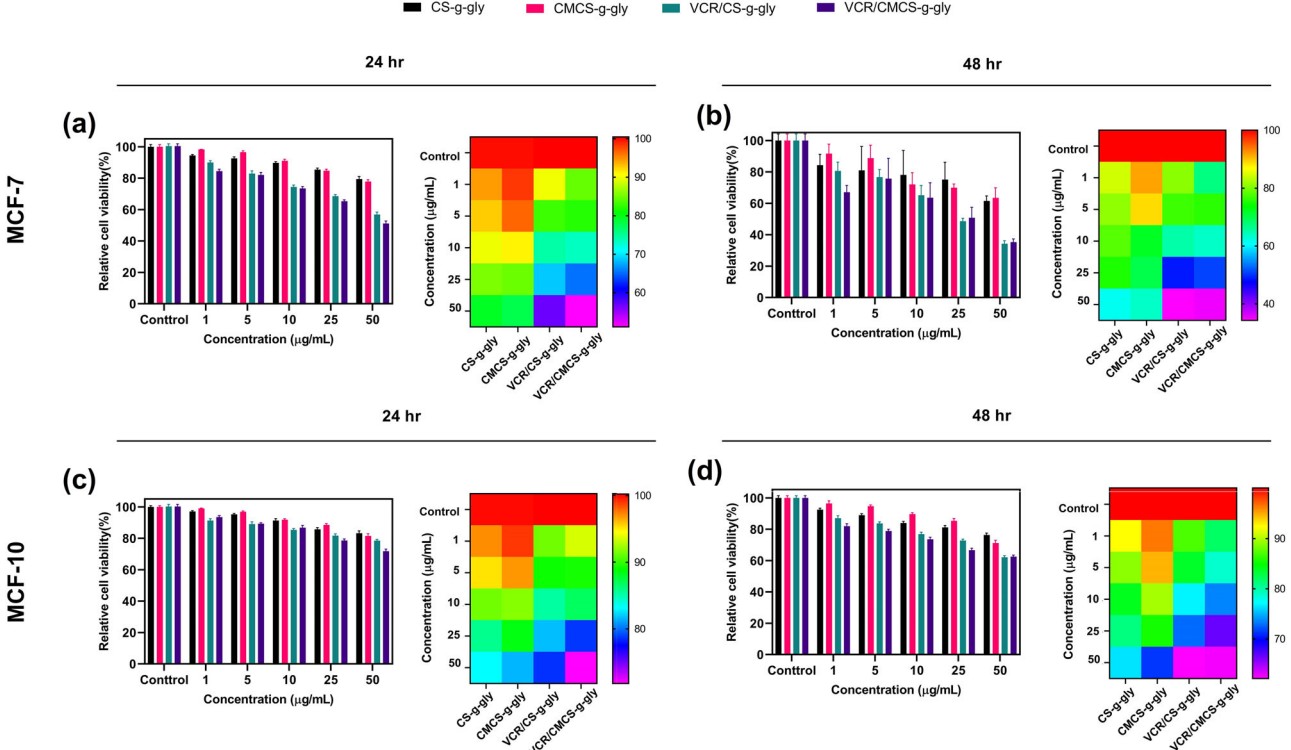

**Fig. 11 Comparison of cytotoxic effect. a, b** Cs-g-gly, CMCS-g-gly, VCR/Cs-g-gly and VCR/CMCS-g-gly against MCF-7 cell lines and (**c, d**) Cs-g-gly, CMCS-g-gly, VCR/Cs-g-gly and VCR/CMCS-g-gly against MCF-10 cell lines, after 24 and 48 h incubation. Each experiment was performed and replicated three times, and the results were displayed using error bars to depict the variance.

induce apoptosis in vincristine-treated MCF-7 cells over 48 h. It could be seen that the cells underwent primary apoptosis and late apoptosis after 48 h, which were affected by the vincristine anti-cancer drug-loaded hydrogels. The apoptosis rate in CS-g-gly loaded with vincristine at $IC_{50} = 22.49$ and CMCS-g-gly loaded with vincristine at $IC_{50} = 24.43$ after 48 h was 52.54 and 46.13%, respectively. Relative frequencies of necrotic and apoptotic cells in culture medium against MCF-7 cell line using Annexin-V/-FITC/ PI Flow cytometry method showed that apoptosis was a major factor in inhibiting cell survival compared with necrosis in hydrogels loaded vincristine anti-cancer drug. A comparision of the present work with the antiapoptotic results in previous studies are represented in Table S4.

## Conclusion

Multi-responsive hydrogels are a kind of exciting material in biomedical research. In our work, two types of chitosan-based and carboxymethyl chitosan-based thermo-sensitive biodegradable, cost-effective, non-toxic, and highly swellable hydrogels were synthesized by a novel method. CS-g-gly and CMCS-g-gly hydrogels were characterized using FT-IR, [1]HNMR, FE-SEM, AFM, TGA, DLS, and zeta potential. CS-g-gly and CMCS-g-gly exhibited high swelling ratios in acidic environments, making them suitable candidates for systemic drug delivery. Vincristine, a hydrophobic chemotherapeutic, was successfully encapsulated and released from the shear-thinning, bio-based nano hydrogel. VCR/CS-g-gly and VCR/CMCS-g-gly showed encapsulation efficiency between 72.28% and 89.97%, and 56.97% and 71.91%, respectively, which could be explained by effective interactions involving covalent bonds and nonvalent bonds. The in vitro release profiles showed the CS-g-gly and CMCS-g-gly hydrogels provided pH-sensitive and sustained drug release, and 82.27% and 75.50% of the VCR in pH = 5 and 25.83% and 5.94% of the

VCR in pH = 7.4 were released from the CS-g-gly and CMCS-g-gly hydrogels after 120 h. The chitosan-based thermo-sensitive gel can be a promising platform for drug delivery systems. MTT assay of CS-g-gly and CMCS-g-gly hydrogels showed no considerable cytotoxicity on MCF-7 and MCF-10 cell lines, indicating their potential for biomedical applications.

## Materials and methods

**Materials**. Chitosan (MW 200,000 (medium molecular weight) and 98% deacetylation), glacial acetic acid, sodium hydroxide, formaldehyde 37 weight percent in water, ammonia 25%, hydrochloric acid 37%, glycerol (gly), ethanol and isopropanol were obtained from Merck (Germany). Vincristine sulfate (VCR) was provided by Godeon Richter plc (Hungary). The MCF-7 and MCF-10 cell lines were obtained from the Cell Bank of Iran's Pasteur Institute. Bromide of 3-(4,5-dimethylthiazol-2-yl)-2,5-diphenyltetrazolium (MTT) was purchased from Sigma-Aldrich (USA). Gibco supplied phosphate-buffered saline (PBS) and Dulbecco's Modified Eagle Medium (DMEM) (USA).

## Preparation of hydrogels

*Preparation of carboxymethyl chitosan (CMCS).* With some modifications to the previous development, we could zsynthesize carboxymethyl chitosan (CMCS)[23]. Briefly, 50 mL of isopropyl alcohol was mixed with 2 g of chitosan (CS) using a magnetic stirrer and let to stir at room temperature for two hours. After transferring the suspension to a 500 mL round bottom flask, 80 mL of aqueous NaOH solution (60%) was added, and the mixture was refluxed at 85 °C for 4 h. Then, 60% w/v monochloroacetic acid solution (100 mL) was added during a 10-minutes period in 5 equal portions. For another 8 h, the mixture was heated to 65 °C while stirring. After that, HCl solution was used to zneutralize the reaction mixture (4 M). After filtering out the insoluble material, methanol precipitated the yielding CMCS. It was then filtered, rinsed many times using a mixture of $CH_3OH/ H_2O$ (1:1), and dried under vacuum.

*Synthesis of CS-g-gly and CMCS-g-gly hydrogels.* To prepare CS-g-gly, 1500 mg of CS was dissolved in 1% (v/v) acetic acid and solution was stirred for 8 h at ambient temperature to obtain a clear chitosan solution. Then 5 mL of 37% formaldehyde was added dropwise, which was stirred for one hour at ambient temperature. 500 mg of glycerol was added dropwise to the solution of chitosan used as a crosslinking agent, followed by stirring with a magnetic stirrer for 6 h. Then, 20 mL of 25% ammonia solution was added to form a hydrogel. The formed hydrogel was

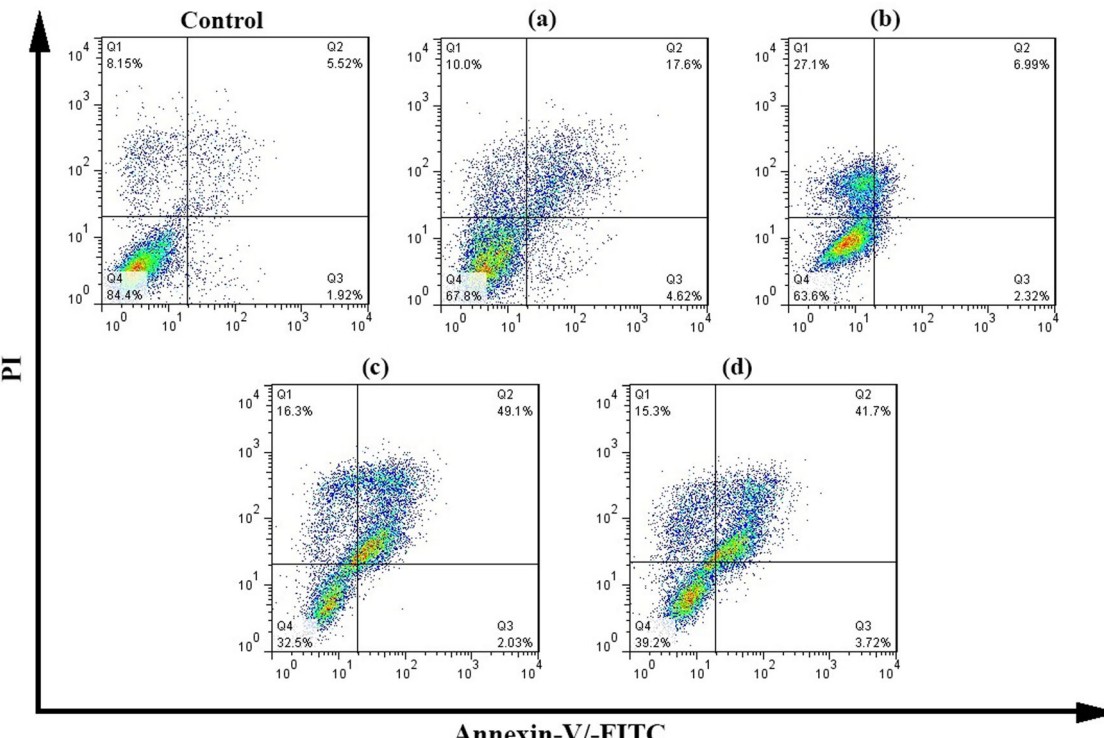

**Fig. 12 Annexin-V/-FITC/PI Flow cytometry analysis of MCF-7 breast cancer cells treated with the prepared hydrogels. a** CS-g-gly, (**b**) CMCS-g-gly, (**c**) CS-g-gly loaded with vincristine, and (**d**) CMCS-g-gly loaded for 48 h. The lower left (Q4) (Annexin−PI−) and the lower right (Q3) (Annexin+PI−) quadrants represent viable cells and early apoptotic cells. The upper right region (Q2) (annexin+PI+) and the upper left (Q1) (annexin− PI+) represent late apoptotic cells and necrotic cells.

rinsed with ethanol to remove any unbonded glycerol and other unreacted material and was filtered via a vacuum pump. Then, it was placed in a vacuum oven to dry (temperature 50 °C). After drying, pulverize the product using a ball mill (Frequency = 25 Hz) for 10 min to be used for further experiments. The same procedure was used for the preparation of CMCS-g-gly, but in place of CS, CMCS was used.

**In vitro drug loading and drug release study**. The anti-cancer drug was encapsulated in the hydrogels, as shown in Table S2. VCR was loaded into the hydrogels for 24 h at ambient temperature and kept in the dark. After adding different amounts of sodium tripolyphosphate (STPP) to the hydrogels, they were centrifuged at 11000 rpm for 30 min. To assess the percentage of entrapment efficiency (EE%), non-entrapped VCR was separated from the entrapped drug in hydrogel networks. A UV-Visible light spectrophotometer was utilized to measure the amount of free drug at 296 nm (UV-1700 PharmaSpec, Shimadzu, Kyoto, Japan). The following equation was employed to determine the EE%[60].

$$EE\% = \left(\text{Total VCR in hydrogel batch} - \text{Free VCR}\right)/\text{total VCR in hydrogel batch} \times 100 \tag{1}$$

The in vitro drug release of CS-g-gly and CMCS-g-gly was investigated with the membrane diffusion technique[60]. Following the determination of the EE%, the pellets were dried and placed in a dialysis bag (cellulose membrane, cut off 12 kD). It was submerged in 50 mL of PBS (with pH values of 5 and 7.4) under magnetic stirring (37 °C, 150 rpm) for 120 h. At certain intervals, 2 mL of the release medium was removed, and the same quantity of fresh PBS was replenished. The drug released at predetermined time intervals were measured at 296 nm by spectrophotometry. Finally, the cumulative drug release percentage was calculated using Eq. 2. The calibration curve equation was also obtained using UV-vis spectrophotometry (at 296 nm wavelength) and by investigating the different concentrations (serial dilutions) of the standard VCR using Excel software.

$$Q = \frac{C_n \times V_0 + V_i \sum_{i=1}^{n-1} C_i}{m} \times 100\% \tag{2}$$

Where $Q$ represent the cumulative drug release (%), $C_n$ indicates the mass concentration of drug released, $V_0$ is the total volume of PBS, $V_i$ is the volume removed per time, $C_i$ is the concentration of drug in the volume removed per time, and $m$ represent total drug loading. The drug release data was used to understand and determine the mechanism and kinetics of VCR release. For data analysis, various kinetic models such as zero-degree, first-degree, Higuchi, and Korsmeyer-Pepas models were investigated and reported in Table S3.

**Characterization of hydrogels**. Avance Bruker DRX-500 spectrometer was used to obtain 1H NMR spectra. The multiplicity of magnetic resonance nuclei is defined as singular (s), binary (d), triplet (t), quaternary (q), and multiple (m). Deuterium (D₂O) was utilized for the solvent, and the signal from the solvent was used for internal calibration (D₂O: δ (1H) = 4.79 ppm). Fourier transform infrared (FTIR) spectra were collected using a FT-IR spectrometer Agilent Cary 630 (Agilent Technologies/EUA). Field emission Scanning Electron Microscopy (SEM) (Topcon/Singapore) was used to study the superficial morphologies of the CS-g-gly and CMCS-g-gly with and without STTP. The samples were coated with a thin 16 nm gold film (Bal-tec/USA). Dynamic and electrophoretic light scattering from a Zetasizer device was used to calculate the mean particle size, size distribution, and zeta potential of hydrogel (Zetasizer Nano ZS, Malvern Instruments, Worcestershire, UK). TGA analyses were performed on a STA 449F3 Calorimeter (Netzsch/Germany) (Model TGA 50, Japan). This calorimetric experiment was carried out at a heating rate of 10 °C.min$^{-1}$ under a nitrogen atmosphere over the following 25–600 °C.

**In vitro swelling of hydrogels**. To determine swelling ratio (SR) of hydrogels, 200 mg tablets were made from each sample (at a pressure of 5 tons). The samples were then placed in 50 mL of buffer solution for 24 h in order to test their swelling behavior at various pH values (1.2, 5, and 7.4) and at various temperatures (5, 25, 37, and 55 °C). The swelling ratio and the equilibrium water content (EWC %) of hydrogels were determined using Eqs. 3 and 4, where $W_s$ and $W_d$ are the weights of the swollen and dried gels, respectively ($N = 3$ for each data point).

$$\text{Swelling ratio} (SR) = (Ws - Wd)/Wd \tag{3}$$

$$\text{Equilibrium water content} (EWC \%) = (Ws - Wd)/Ws \times 100 \tag{4}$$

**MTT assay**. Cell viability was assessed against MCF7 and MCF10 cell lines by 3-(4,5-dimethylthiazol-2-yl)-2,5-diphenyltetrazolium bromide (MTT) assay. In this regard, cells were cultured at a density of $1 \times 10^4$ cells/well in RPMI-1640 medium containing 10% heat-inactivated FBS and 1% penicillin and streptomycin. Then, they were incubated with 5% CO₂ in a humidified atmosphere. It was followed by replacing the medium with fresh medium containing serially diluted hydrogel samples and free VCR after 48 h. In the next step, a solution of MTT dye in PBS containing 5 mg/mL was added to each well. After incubating for 4 h with the medium and MTT, the formazan crystals were dissolved with 200 mL of DMSO. After placing the plate at 37 °C for 10 min, the measurement was performed. A

microplate reader measured optical density (OD) at 540 nm.

$$Cell\ viability = OD\ of\ test / OD\ of\ control \times 100 \tag{5}$$

**Flow cytometry**. A flow cytometry analysis was used to determine the percentage of apoptotic MCF-7 cells. For 48 h, the cultivated cells were treated with CMCS-g-gly, CS-g-gly, and drug-loaded hydrogels. The Apoptosis and Necrosis Quantitation Kit was employed for dyeing green the cells that were apoptotic and red those that were necrotic in order to carry out flow cytometry analysis. After two PBS rinses, the cells were suspended in 1X binding buffer ($5 \times 10^5$ cells/mL). The cells were stained with annexin V-FITC (green fluorescence) and propidium iodide (red fluorescence) successively and examined using a tabletop flow cytometer (FACSCalibur, D Biosciences, Franklin Lakes NJ, USA).

**Statistical analysis**. All results were presented as mean ± standard deviation (SD). The two-way ANOVA and independent t-test were used to examine the statistical significance of differences; a $P$ value $< 0.01$ was considered statistically significant. Statistical significance was pre-set at $\alpha = 0.05$ for all analyses.

**Reporting summary**. Further information on research design is available in the Nature Portfolio Reporting Summary linked to this article.

## Data availability
The $^1$HNMR data is available at supplementary information. All other relevant data are available from the authors upon request.

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

## Author contributions

Conceptualization: BFF, MO, and MRN; Data curation: BFF; Formal analysis: BFF; Funding acquisition: all authors; Investigation: BFF; Methodology: BFF and MO; Project administration: SJ, MRN; Resources; Software; Validation; Visualization: BFF; Writing—original draft: BFF; Writing—review & editing: SJ, MRN, and BFF.

## Competing interests

The authors declare no competing interests.

## Ethical approval

This study has no animal study or human subjects.
