## [Peer Review File · Communications Chemistry]

Reviewers' comments:

Reviewer #1 (Remarks to the Author):

In this study, authors prepared multi-responsive hydrogels for controlled drug release system. I urge the authors to address the following suggestions.

1. There are some grammatical mistakes in manuscript. Please revise the manuscript thoroughly.
2. What are MCF-7 and MCF-10 used in abstract. Don't use such abbreviation in abstract.
3. Please do not use apostrophes in manuscript. Make the sentences simple.
4. What is new and novel in the study beyond the already reported in literature.
5. Following papers can be cited to explain the rationale of the study. International Journal of Biological Macromolecules, 162 (2020) 175-187. <https://doi.org/10.1016/j.ijbiomac.2020.06.133>, International Journal of Biological Macromolecules, 164 (2020) 4370-4380. <https://doi.org/10.1016/j.ijbiomac.2020.09.031>, Polymers for Advanced Technologies, 30 (2019) 2414-2424. <https://doi.org/10.1002/pat.4688>. RSC Advances, 9 (2019) 31078-31091. <https://doi.org/10.1039/C9RA05025F>
6. Why authors chosen the parameters of temperature of 5 and 55 C for swelling analysis?
7. IN figure 3, replace the unit of transmittance as "a.u".
8. In Figure 7, y-axis legend should be weight (%). Why weight is greater than 100% in figure 7d?
9. As shown in Figure 8, swelling ratio at pH 1.2 is less than at pH 5 and 7.2. But authors written it is greater?
10. With increase in the percentage of STPP, how the entrapment efficiency is decreased?
11. please cite some latest papers of this journal.
12. Authors have reported this study for biomedical application but didn't perform antibacterial study. Please include antibacterial analysis of prepared hydrogels.

Reviewer #2 (Remarks to the Author):

Comments on Manuscript for Communications Chemistry

Title: " Multi-Responsive Chitosan-Based Hydrogels for Controlled Release of Vincristine: Design, Characterization, and Safety Evaluation " Correspondence Author: Bahareh Farasati Fara, Mohsen Omrania, M. Reza Naimi-Jamala*, Shahrzad Javanshirb*

This is a good manuscript reporting two well designed chitosan-based drug carried hydrogels. Two chitosan-based hydrogels were synthesized and were served as drug carriers in order to solve the problem when using Vincristine for treating cancer. Rather high encapsulation efficiency (72.28-89.97%, and 56.97-71.91%) was shown comparing to some existing materials. The work is complete and various data were well collected. Several crosslinker (ammonium, glycerol, STPP) were used in this work, however little detail was discussed. Finally, it would be better if the biodegradability of two hydrogels was evaluated. It is acceptable for publication with minor revision.

Some questions and comments were listed below:

1. Various hydrogels were used as drug carrier however none of one was mentioned. Further, is there any advantage comparing these drug-hydrogel systems?
2. Non-toxic was mentioned in the paper, however formaldehyde was used as material. The amount of remaining formaldehyde might be determined.
3. Why chitosan (CS) and carboxymethyl chitosan (CMCS) were studied? The reason (logic) was not mentioned.

2.2 Scheme is suggested.

2.5 What is the benefit of a high swelling ratio?

3.1 Is there any evidence of interaction between gel and Vincristine?

3.2.4 The conditions of experiments were missing.

Fig. 11 The annotation of the value is needed.

Table 4. A short conclusion is suggested.

Reviewers' comments:

Reviewer #1 (Remarks to the Author):

In this study, authors prepared multi-responsive hydrogels for controlled drug release system. I urge the authors to address the following suggestions.

1. There are some grammatical mistakes in manuscript. Please revise the manuscript thoroughly.

Response: Thank you for your comment. The typos and grammatical errors have been edited.

2. What are MCF-7 and MCF-10 used in abstract. Don't use such abbreviation in abstract.

Response: Thank you for your comment. MCF-7 and MCF-10 are cell lines. We have edited the abstract to make it more clear.

3. Please do not use apostrophes in manuscript. Make the sentences simple.

Response: Thank you for your comment. These grammatical errors have been edited.

4. What is new and novel in the study beyond the already reported in literature.

Response: Thank you for your comment. This hydrogel were synthesized using cost-effective cross linkers and biodegradable polysaccharide. This hydrogel has superadsorbent of water which can be used in other applications too. Also the minimal toxicity for normal cells and efficiency for inducing apoptosis in cancer cell lines would be a highlight point of this work.

5. Following papers can be cited to explain the rationale of the study. International Journal of Biological Macromolecules, 162 (2020) 175-

187. <https://doi.org/10.1016/j.ijbiomac.2020.06.133>, International Journal of Biological Macromolecules, 164 (2020) 4370-4380. <https://doi.org/10.1016/j.ijbiomac.2020.09.031>,

Polymers for Advanced Technologies, 30 (2019) 2414-

2424. <https://doi.org/10.1002/pat.4688>. RSC Advances, 9 (2019) 31078-

31091. <https://doi.org/10.1039/C9RA05025F>

Response: Thank you for your suggestions. All these papers have been cited in the manuscript.

6. Why authors chosen the parameters of temperature of 5 and 55 C for swelling analysis?

Response: Thank you for your comment. These parameters of temperature have been selected to better evaluate the thermoresponsive characteristic of synthesized hydrogel.

7. IN figure 3, replace the unit of transmittance as "a.u".

Response: Thank you for your comment. This figure has been edited.

8. In Figure 7, y-axis legend should be weight (%). Why weight is greater than 100% in figure 7d?

Response: Thank you for your comment. The TGA evaluation in nitrogen (or inert gas), may result to be absorbed or due to not calibrating and degassing the device. Therefore, weight may be greater than 100% in some cases.

9. As shown in Figure 8, swelling ratio at pH 1.2 is less than at pH 5 and 7.2. But authors written it is greater?

Response: Thank you for mentioning this point. This error has been edited.

10. With increase in the percentage of STPP, how the entrapment efficiency is decreased?

Response: The entrapment efficiency was directly proportional to the concentration of STPP as it increased significantly by increasing the STPP concentration up-to a certain level, which corresponds optimized ionic gelation of the components. Above this value, entrapment efficiency decreased which can be attributed to the shells becoming thicker with the STPP concentration and relative competition between the Vincristine and STPP within the nanoparticles. [Ref: doi: 10.2147/IJN.S232350.]

11. please cite some latest papers of this journal.

Response: Thank you for your comment. New papers has been added from this journal.

12. Authors have reported this study for biomedical application but didn't perform antibacterial study. Please include antibacterial analysis of prepared hydrogels.

Response: Thank you for your comment. This hydrogel can be used in biomedical application as well when it can be loaded by antibiotics and other antimicrobial agents. However, in this manuscript the anticancer activity through chemotherapeutic delivery has been investigated. Therefore antibacterial test has not been conducted in this work. But to benefit from your valuable comment, in our next manuscript for antibiotic delivery in biomedical field, this test

will be conducted.

Reviewer #2 (Remarks to the Author):

Comments on Manuscript for Communications Chemistry

Title: " Multi-Responsive Chitosan-Based Hydrogels for Controlled Release of Vincristine: Design, Characterization, and Safety Evaluation " Correspondence Author: Bahareh Farasati Fara, Mohsen Omrania, M. Reza Naimi-Jamala*, Shahrzad Javanshirb*

This is a good manuscript reporting two well designed chitosan-based drug carried hydrogels. Two chitosan-based hydrogels were synthesized and were served as drug carriers in order to solve the problem when using Vincristine for treating cancer. Rather high encapsulation efficiency (72.28-89.97%, and 56.97-71.91%) was shown comparing to some existing materials. The work is complete and various data were well collected. Several crosslinker (ammonium, glycerol, STPP) were used in this work, however little detail was discussed. Finally, it would be better if the biodegradability of two hydrogels was evaluated. It is acceptable for publication with minor revision.

Some questions and comments were listed below:

1. Various hydrogels were used as drug carrier however none of one was mentioned. Further, is there any advantage comparing these drug-hydrogel systems?

Response: Thank you for your comment. Two chitosan-based hydrogels have been synthesized in this work to evaluate the relationship between the chemical structures of hydrogels with some parameters, such as temperature, pH, EE, toxicity and etc. which have been detailed in manuscript as well.

2. Non-toxic was mentioned in the paper, however formaldehyde was used as material. The amount of remaining formaldehyde might be determined.

Response: Thank you for your valuable point. This question has been our concern from the beginning of the work. After some evaluation on the final synthesized hydrogel, it has been shown that there is no evidence of remaining formaldehyde in final synthesized hydrogel. This is

due to several (5-6 times) washing cycles with absolute ethanol and using the vacuum pump for each cycle to evacuate any residual and unreacted substance such as formaldehyde.

3. Why chitosan (CS) and carboxymethyl chitosan (CMCS) were studied? The reason (logic) was not mentioned.

Response: Thank you for your comment. Chitosan (CS) derivatives are commonly used due to their unique properties like biocompatibility, mucoadhesion, non-toxicity and capability to form gels. Carboxymethyl chitosan (CMCS) has appeared as an auspicious biopolymer for the development of new drug delivery systems. CMCS an amphiprotic ether derived from chitosan, exhibit improved aqueous solubility, outstanding biocompatibility, biodegradability, osteogenesis ability and numerous other outstanding physicochemical and biological properties. More stupendously, it can load hydrophobic drugs and displays strong bioactivity which highlight its suitability and extensive usage for preparing different drug delivery systems. Chitosan as a biodegradable polymer which is mostly use as hydrogels. Chitosan has many advantages such as low-price in comparison to many costly polymers, being pH-responsive which is highly important in drug delivery and tumor therapy.

2.2 Scheme is suggested.

Response: Thank you for your comment. It is unclear which scheme the referee is talking about.

2.5 What is the benefit of a high swelling ratio?

Response: Thank you for your comment. From a physicochemical point of view, the swelling ability of hydrogels is a fundamental property that should influence the drug release rates. The higher swelling ratio characteristics of this structure lead to effective control of the drug release.

3.1 Is there any evidence of interaction between gel and Vincristine?

Response: Thank you for your comment. The interaction of hydrogel and Vincristine is depicted in Figure 1. The EE and Loading percentage could be also evidence for interaction of hydrogels and vincristine which is in a good agreement with the swelling of hydrogels and the structures suggested in Figure 1.

3.2.4 The conditions of experiments were missing.

Response: Thank you for your comment. The condition of experiment has been checked and add detailed where were possible.

Fig. 11 The annotation of the value is needed.

Response: Thank you for your comment. The annotation of the value is given in the top of the figure.

Table 4. A short conclusion is suggested.

Response: Thank you for your comment. This Table has been concluded in introduction, discussion, and conclusion part.

REVIEWERS' COMMENTS:

Reviewer #1 (Remarks to the Author):

Authors have adequately address the suggested comments.